# Machine learning models exploring characteristic single-nucleotide signatures in yellow fever virus

Álvaro Salgado[1]☯*, Raquel C. de Melo-Minardi[2]☯, Marta Giovanetti[1,3]☯,
Adriano Veloso[2]☯, Francielly Morais-Rodrigues[1], Talita Adelino[4], Ronaldo de Jesus[5],
Stephane Tosta[1], Vasco Azevedo[1], José Lourenco[6]*, Luiz Carlos J. Alcantara[1,3]*

1 Laboratório de Genética Celular e Molecular, Instituto de Ciências Biológicas, Universidade Federal de Minas Gerais, Belo Horizonte, Minas Gerais, Brazil, 2 Departamento de Ciência da Computação, Instituto de Ciências Exatas, Universidade Federal de Minas Gerais, Belo Horizonte, Minas Gerais, Brazil, 3 Laboratório de Flavivírus, Instituto Oswaldo Cruz, Fundação Oswaldo Cruz, Rio de Janeiro, Brazil, 4 Laboratório Central de Saúde Pública, Fundação Ezequiel Dias, Belo Horizonte, Minas Gerais, Brazil, 5 Coordenação Geral dos Laboratórios de Saúde Pública, Secretaria de Vigilância em Saúde, Ministério da Saúde, Brasília, DF, Brazil, 6 Department of Zoology, University of Oxford, Oxford, United Kingdom

☯ These authors contributed equally to this work.
* alvarosalgado@ufmg.br (AS); luiz.alcantara@ioc.fiocruz.br (LCJA); jose.lourenco@zoo.ox.ac.uk (JL)

**Data Availability Statement:** Data was retrieved from open access sources and repositories, available at: http://www.nature.com/articles/s41598-019-56650-1 https://science.sciencemag.

## Abstract

Yellow fever virus (YFV) is the agent of the most severe mosquito-borne disease in the tropics. Recently, Brazil suffered major YFV outbreaks with a high fatality rate affecting areas where the virus has not been reported for decades, consisting of urban areas where a large number of unvaccinated people live. We developed a machine learning framework combining three different algorithms (XGBoost, random forest and regularized logistic regression) to analyze YFV genomic sequences. This method was applied to 56 YFV sequences from human infections and 27 from non-human primate (NHPs) infections to investigate the presence of genetic signatures possibly related to disease severity (in human related sequences) and differences in PCR cycle threshold (Ct) values (in NHP related sequences). Our analyses reveal four non-synonymous single nucleotide variations (SNVs) on sequences from human infections, in proteins NS3 (E614D), NS4a (I69V), NS5 (R727G, V643A) and six non-synonymous SNVs on NHP sequences, in proteins E (L385F), NS1 (A171V), NS3 (I184V) and NS5 (N11S, I374V, E641D). We performed comparative protein structural analysis on these SNVs, describing possible impacts on protein function. Despite the fact that the dataset is limited in size and that this study does not consider virus-host interactions, our work highlights the use of machine learning as a versatile and fast initial approach to genomic data exploration.

## Introduction

Yellow fever (YF) is an acute viral hemorrhagic disease endemic in tropical areas of Africa and Latin America. The causative agent, yellow fever virus (YFV), represents the prototypical

org/content/361/6405/894 https://jvi.asm.org/
content/94/1/e01623-19 https://dx.plos.org/10.
1371/journal.pntd.0008405 https://dx.plos.org/10.
1371/journal.ppat.1008699 Code is available at
GitHub (https://github.com/alvarosalgado/yfv_
code).

**Funding:** A.S. was supported by Decit, SCTIE,
Brazilian Ministry of Health, Conselho Nacional de
Desenvolvimento Científico - CNPq - (Grants
440685/2016-8 and 440856/2016-7) https://www.
gov.br/cnpq/pt-br; Coordenação de
Aperfeiçoamento de Pessoal de Nível Superior -
CAPES (Grants 88887.130716/2016-00,
88881.130825/2016-00 and 88887.130823/2016-
00) https://www.gov.br/capes/pt-br; The European
Union's Horizon 2020 Research and Innovation
Programme under ZIKAlliance Grant Agreement
No. 734548 - https://zikalliance.tghn.org. All other
authors received no specific funding for this work.
The funders had no role in study design, data
collection and analysis, decision to publish, or
preparation of the manuscript.

**Competing interests:** The authors have declared
that no competing interests exist.

member of the genus *Flavivirus* (family *Flaviviridae*), consisting of a single-stranded, positive-sense RNA virus, with a genome about 11,000 kb and a single open-reading frame of 10,233 nucleotides [1, 2]. Disease varies from nonspecific febrile illness to a fatal hemorrhagic fever. Symptoms usually appear after an incubation period of three to six days following the bite of an infected mosquito, with a period of infection lasting several days [2–4]. The World Health Organization (WHO) reports case fatality rates in the order of 15 to 50% [5]. Vaccination remains the most effective YF prevention method, providing lifetime immunity in up to 99% of vaccinated people [6]. Nevertheless, the burden of YF is estimated to be between 84,000 to 170,000 severe cases and 29,000 to 60,000 deaths annually [7, 8], while an estimated 35 million people remain unvaccinated in areas at risk in Brazil only [9].

YFV spreads in two different cycles: sylvatic and urban. The sylvatic transmission cycle occurs in forested areas, where the virus is endemically transmitted between several non-human primate (NHP) species. The urban transmission cycle occurs when the virus is introduced into human populations with high density and urban-dwelling mosquitoes (mainly *Aedes aegypti*) [3]. Urban cycles of YFV transmission have been eradicated in Brazil since 1942 due to vaccination and vector control campaigns [10–13].

In the last decade in Brazil, however, human and NHP epizootic YF cases have been notified at places beyond the limits of regions previously considered (sylvatic) endemic for the virus [14–17]. The severe impact of these recent outbreaks can be measured, in part, by its fatality rate at around 34%, higher than the general rate estimated by Monath and colleagues [4], motivating the inquiry as to what could be the possible factors contributing to such a high fatality rate, and if YFV genetic signatures could be among those factors.

Additionally, important findings in recent epidemics [11, 18] show a significant difference in the distribution of NHP Ct values, in which *Callithrix* spp. exhibit generally higher Ct values than other NHP species, do not develop fatal YFV infections similar to those reported in humans and can persist for longer, thus increasing the infectious period. The latter can be an essential factor in igniting an urban cycle of transmission, mainly due to the genus' proximity to densely urbanized areas.

On this respect, genomic and epidemiological monitoring have become an integral part of the national (Brazil) and international response to emerging and ongoing epidemics of viral infectious diseases, allowing the availability of a large amount of genomic data [19–23].

In genomic and epidemiological monitoring analysis, machine learning (ML) approaches are usually applied [24, 25], as described in works that analyze the effectiveness of large scale genome-wide association studies (GWAS), due to their capability to computationally model the relationship between combinations of single nucleotide variants, other genetic variations and environmental factors with observed outcomes [26–28].

We curated two different datasets of YFV genomes, one from human cases and the other from NHP cases. After data curation, the human dataset contained 56 YFV sequences, with 40 sequences related to infections leading to severe outcomes or death, and 16 sequences related to cases with no severe outcome. We also gathered an NHP (*Callithrix* spp.) dataset, that after curation contained 27 sequences, of which 21 were related to low Ct values ($< 20$) and 6 were related to high Ct values ($> = 20$).

We applied three different ML models to each dataset, to guarantee robustness of the ML analysis [29]. We then analyzed the models using SHAP (SHapley Additive explanation) [30–32] to highlight genetic signatures. The possible biological impacts of these signatures were investigated and discussed by means of in-silico protein structural analysis coupled with literature review.

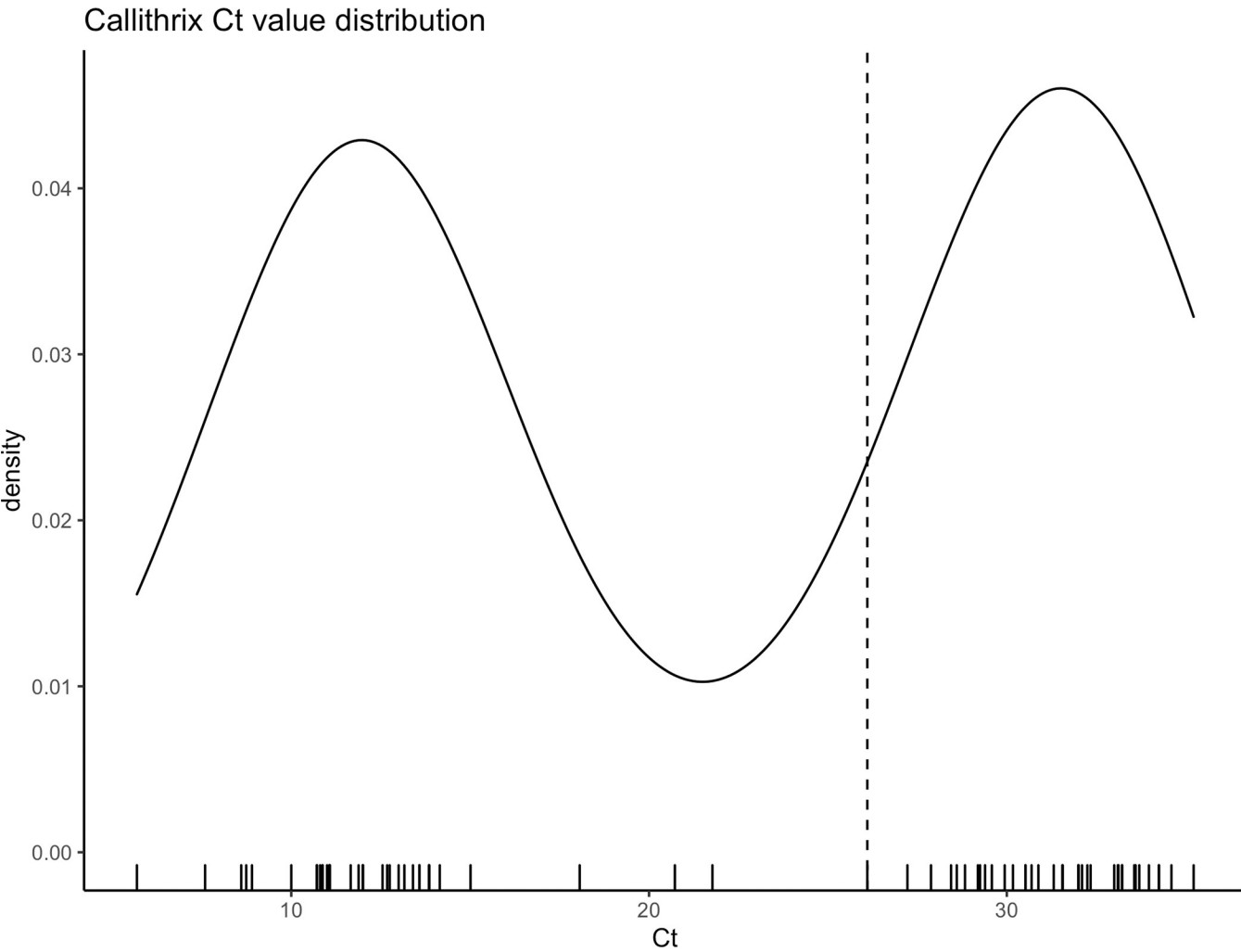

**Fig 1. Ct values associated with *Callithrix* spp. sequences.** The median value is shown by a dashed line. Hartigan's dip test of unimodality indicates bimodal distribution (p < .001).

## Results

### Non-human primates Ct value statistical analysis

Fig 1 shows the distribution of cycle threshold (Ct) values from *Callithrix* spp. sequences, with two distinct clusters roughly around 12 and 30, with a median value of 26.1. The result of Hartigan's dip test of unimodality [33] rejected the null hypothesis of a unimodal distribution for Ct values ($p < .001$), which indicate the existence of two groups of *Callithrix* spp. Ct values.

### Machine learning models' performance

Fig 2 shows the confusion matrices for the machine learning models applied. For the human dataset, the XGBoost classification model correctly classified 16 serious/death and 5 not serious/not death cases, out of a total of 28 instances, achieving an accuracy of 75% on the test set, with F-1 scores of 0.59 and 0.82 for classes 0 (not severe/not death) and 1 (severe/death), respectively. The random forest model correctly classified 16 serious/death and 7 not serious/not death cases, out of a total of 28 instances, achieving an accuracy of 82% on the test set, with

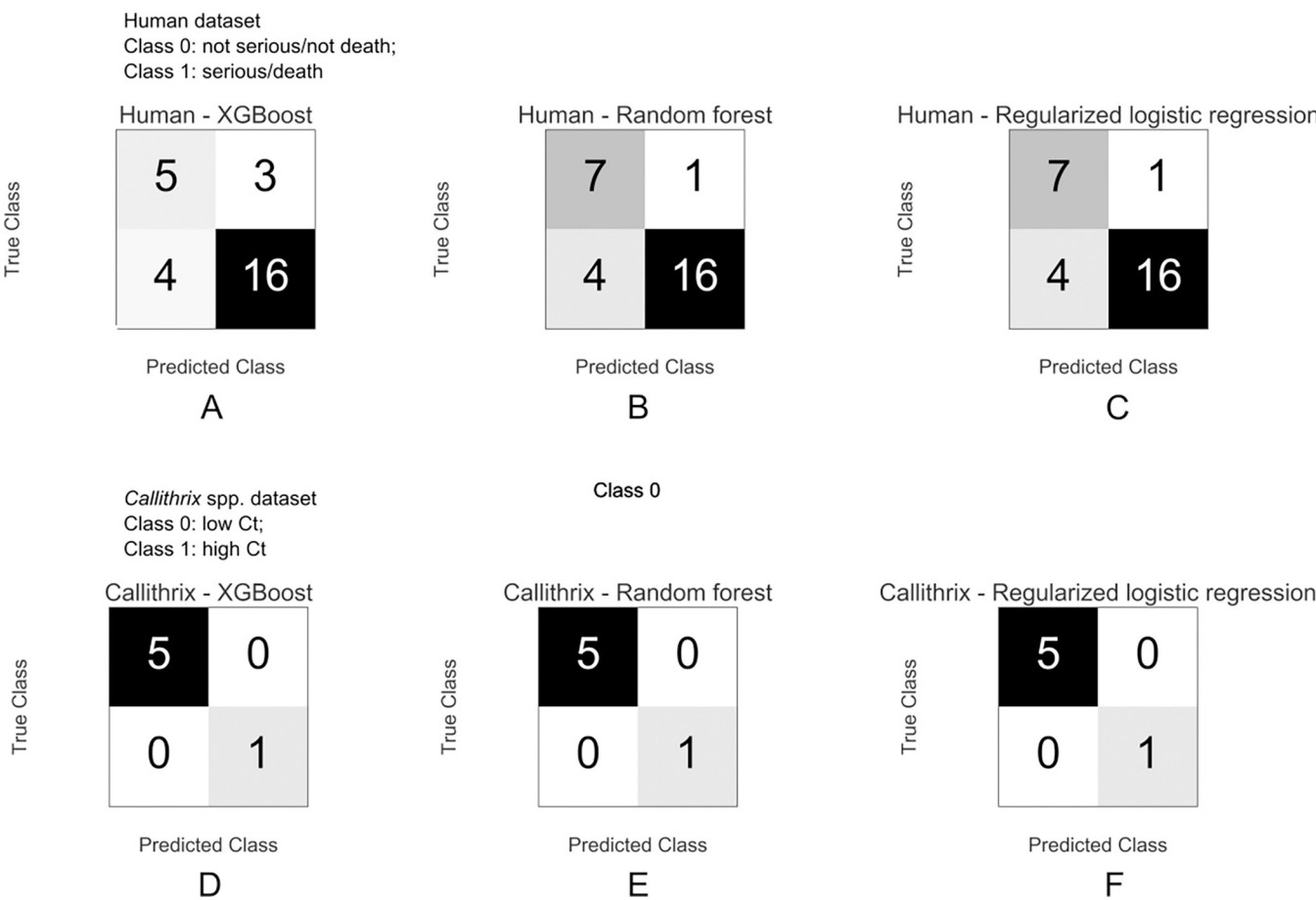

**Fig 2. Confusion matrices.** Each box contains one confusion matrix, which measures the performance of different machine learning algorithms over different datasets. For each matrix, the rows represent instances in the actual class, while the columns represent instances in the predicted class. Instances found along the diagonal were correctly classified, while those outside the diagonal were misclassified (false positives and false negatives). Top three matrices correspond to human test dataset and the bottom three matrices correspond to Callithrix spp. test dataset, for XGBoost, random forest and regularized logistic regression respectively.

F-1 scores of 0.74 and 0.86 for classes 0 and 1 respectively. The modified logistic regression model performance was the same as that of random forest.

For the *Callithrix* spp. dataset, the three models achieved an accuracy of 100% on the test set, correctly predicting 5 low Ct cases and 1 high Ct case, out of 6 instances in the test set, with F-1 scores of 1.00 and 1.00 for classes 0 (low Ct) and 1 (high Ct) respectively.

## YFV genetic signatures

The machine learning methods identified the non-synonymous SNVs shown in Table 1. The table displays each protein where the SNV was found, with nucleotide position on YFV genome, position relative to the protein's sequence, amino acid position relative to the translated protein, reference genome amino acid and corresponding codon, analyzed sequences amino acid variation and corresponding codon and SNV position inside codon (1st, 2nd or 3rd).

The results obtained by the analysis of human YFV sequences highlighted 4 SNV positions that result in the amino acid change, and the results obtained by the analysis of *Callithrix* spp. shows 6 SNV positions that resulted in the amino acid change.

**Table 1. YFV genetic signatures.**

| | | | | | | | | Human dataset |
|---|---|---|---|---|---|---|---|---|
| Protein | nn position | nn position on protein | aa position on protein | aa reference | codon reference | aa variation | codon variation | SNV codon position (1, 2, 3) |
| NS3 | 6412 | 1842 | 614 | E | gaa | D D | gac gat | 3 |
| NS4a | 6644 | 205 | 69 | I | atc | V | gtc | 1 |
| NS5 | 9815 | 2179 | 727 | R | agg | G | ggg | 1 |
| NS5 | 9564 | 1928 | 643 | V | gtt | A | gct | 2 |
| | | | | | | | | NHP dataset |
| Protein | nn position | nn position on protein | aa position on protein | aa reference | codon reference | aa variation | codon variation | SNV codon position (1, 2, 3) |
| NS5 | 8756 | 1120 | 374 | I | atc | V | gtc | 1 |
| NS5 | 9559 | 1923 | 641 | E | gaa | D D | gac gat | 3 |
| NS3 | 5120 | 550 | 184 | I | atc | V | gtc | 1 |
| E | 2126 | 1153 | 385 | L | ctc | F | ttc | 1 |
| NS5 | 7647 | 11 | 4 | N | aat | S | agt | 2 |
| NS1 | 2964 | 512 | 171 | A | gca | V | gta | 2 |

## Protein structural analysis

We performed protein structural analysis for all SNVs indicated in Table 1. The templates, their resolution, the quality of models provided from Swiss-Model [34] and the changes in binding affinity and stability predicted by mCSM-NA [35] are summarized in Table 2.

Fig 3 shows the structural representation of proteins E, NS1, NS3 and NS5, with corresponding SNVs.

**E (envelope) protein.** LEU385 (NHP dataset) is far from the intra-chain binding site and the antibody recognition site (Fig 3A). LEU385 is in the Domain III (DIII) which has an immunoglobulin C domain (IgC-like) presenting a seven-stranded fold and is supposed to contain the receptor-binding site. DIII suffers a rotation and goes closer to the fusion loop (FL), bringing the C-terminal part of DIII (residue 392) close to FL. LEU385 is 20.3 angstroms far from GLU392.

**NS1 protein.** The SNV A171V (NHP dataset) is in close contact with a region called *wing flexible loop* (highlighted in orange in Fig 3B) and it is also a probable glycosylation site [36].

**NS3 protein.** Although there is a crystallographic structure (PDB id: 1yks) of the helicase domain [37] and another containing part of NS3 complexed with NS2B (PDB id 6urv) [38] deposited in the PDB, the referred SNV occurs in the unresolved stretch. The I184V (NHP dataset) is in the linker region (shown in orange in Fig 3C, left). It connects protease and helicase domains and corresponds to sequence KEEGKEELQEIP that encompasses residues between 174 and 185. The SNV E614D (human dataset) occurs in the helicase domain and is located in the RNA binding cleft, shown in red on Fig 3C, right.

**NS4a protein.** Since it has multiple transmembrane hydrophobic segments, structural analysis of NS4a has been unsuccessful and, so far, there is no structure deposited in the PDB. It is still one of the least characterized proteins from YFV. It was not possible to obtain a good structural model since the best template found had a coverage of only 37% and a sequence identity of 25.53%.

**NS5 protein.** There is a recent YFV NS5 structure deposited on PDB (PDB id 6qsn) [39]. The analyzed SNVs (shown in green in Fig 3D) are N11S, the only SNV in the MTase domain; I374V and E641D, both located in the palm subdomain. These three SNVs were found on the NHP dataset. Additionally, SNV V643A, located in the palm, and R727G, in the thumb subdomain, were found in the human dataset. As depicted in Fig 3D, they (green) are located far from the Zn and sulfate ions and the ligand (S-adenosyl-L-homocysteine) (grey). They are not

**Table 2. Protein structural analysis results.**

| | Callithrix spp. Dataset | | | | | | Human dataset | | | |
|---|---|---|---|---|---|---|---|---|---|---|
| Protein | E | NS1 | NS3 | NS5 | | | NS3 | NS4a | NS5 | |
| SNV | L385F | A171V | I184V | N11S | I374V | E641D | E614D | I69V | R727G | V643A |
| Template (PDB ID) | Template (6WI5) (?) | Zika virus NS1 (5K6K) (27) | Dengue virus NS3 (5YV8:A) (31) | Yellow fever virus NS5 (6QSN) (32) | Yellow fever virus NS5 (6QSN) (32) | Yellow fever virus NS5 (6QSN) (32) | Yellow fever virus NS3 (1YKS) (29) | No structure deposited in the PDB | Yellow fever virus NS5 (6QSN) (32) | Yellow fever virus NS5 (6QSN) (32) |
| Resolution | 1.83 Å | 1.89 Å | 2.5 Å | 3.00 Å | 3.00 Å | 3.00 Å | 1.80 Å | - | 3.00 Å | 3.00 Å |
| Coverage (%) | | 100% | 99% | | | | | - | | |
| Sequence identity | | 47.58% | 50.57% | | | | | - | | |
| Localization at protein | Domain III | Wing flexible loop | Linker region | MTase domain | Palm subdomain | Palm subdomain | Helicase domain | - | Thumb subdomain | Palm subdomain |
| Global Model Quality Estimation (GMQE) | | 0.79 | 0.76 | | | | | - | | |
| (Qualitiy mean) QMEAN | | -2.5 | -2.22 | | | | | - | | |
| Predicted change in binding affinity | Target distant from binding sites | | | | ΔΔG = 0.003 Kcal/mol (Increased affinity) | | ΔΔG = 0.025 Kcal/mol (Increased affinity) | - | ΔΔG = -1.545 Kcal/mol (Reduced affinity) | ΔΔG = -0.001 Kcal/mol (Reduced affinity) |
| Predicted change in stability | | | | | -1.908 Kcal/mol (Destabilising) | | -0.254 Kcal/mol (Destabilising) | - | -0.751 Kcal/mol (Destabilising) | -1.884 Kcal/mol (Destabilising) |

close to any important / conserved mentioned residue (yellow) that interact with the nucleic acid. SNV I374V is also present in ZIKV, DENV and WNV. Position E641D varies across other viruses (K, N, R). Position V643A is also not conserved being an insertion, K or N. Position R727G is S, E or T in other flaviviruses.

## Discussion

Emerging and reemerging viruses present a highly complex challenge for the Brazilian public health system. Among them, arboviruses transmitted by mosquitoes are agents capable of causing serious diseases, such as hemorrhagic fevers, encephalitis and meningitis. For these reasons, real-time genomic surveillance is extremely important to guide prevention and control measures, as it allows reconstruction of the origins of epidemics and the estimation of transmission rates at different times and geographic regions, subject to environmental and human factors. In addition, genomic surveillance makes it possible to identify emerging, re-emerging, circulating and co-circulating variants, through viral genetic diversity quantification, making it possible to estimate the likelihood of new outbreaks and/or possible escapes from existing vaccines and treatments. As a result, relevant information is acquired for the design of public health policies, in addition to contributing to the development of vaccines, new drugs and improved serological and molecular diagnostic methods [40, 41].

In this context, Brazil has become a global reference in real-time genomic surveillance, achieving fundamental results in early detection and monitoring of outbreaks. However, the

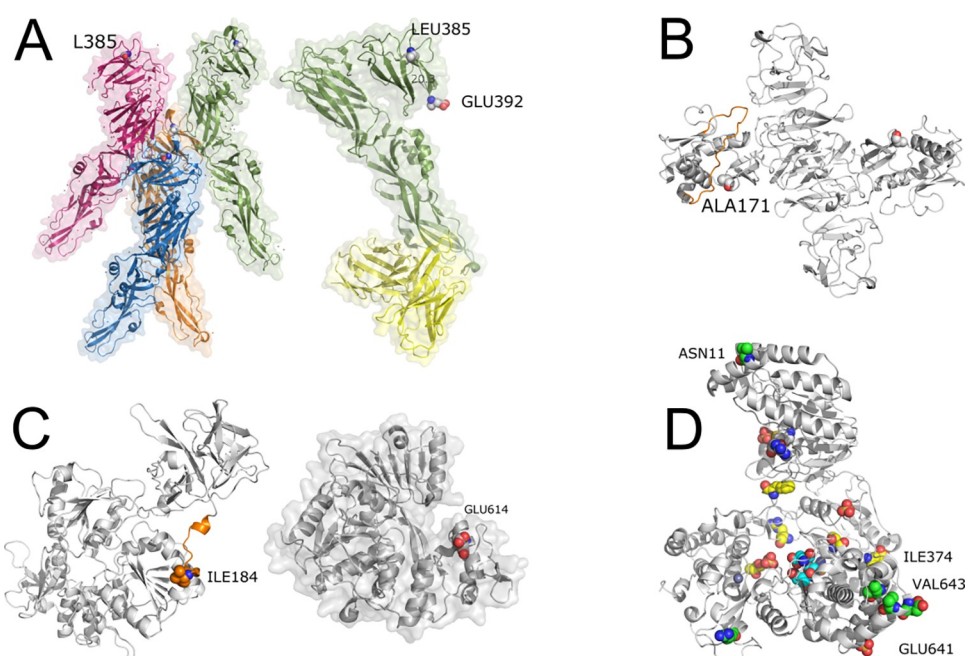

**Fig 3. YFV proteins analysis.** (A) Envelope protein—(A, left) PDB id 6wi5—tretamer of E protein LEU385 presented in spheres. (A, right) PDB id 6ivz—in green, one chain of protein E and in yellow, the light and heavy chains of monoclonal antibody 5A. Note that the L385F SNV is distant from both the binding between protein E chains and the antibody recognition site. (B) NS1 protein. Comparative model built with Swiss-Model and PDB id 5k6k. In orange, we depict the wing flexible loop and in spheres, SNV ALA171. (C) NS3 protein. (C, left) Comparative model built with Swiss-Model and PDB id 5yvu. In orange, we depict the interdomain linker region and in spheres, SNV ILE184. (C, right) GLU614 from PDB id 1yks showing that the SNV E614D is close to the cleft where the DNA binds the NS3 protein. (D) NS5 protein. In green, SNVs. In yellow, important conserved residues across ZIKV, DENV and WNV. In cyan, active site. In grey, ligand S-adenosyl-L-homocysteine. Sulfate and Zn íons are also represented in spheres.

large amount of data produced using next-generation sequencing platforms demands sophisticated analytical approaches, capable of dealing with complex and large datasets, aiming at the extraction of as much information as possible. In this sense, Machine Learning algorithms have been successfully used in Bioinformatics, motivating their application in the search for genetic signatures in arboviruses, associated with phenotypic or epidemiological characteristics in recent outbreaks in Brazil.

In this study, we demonstrate the potential of applying ML approaches on real-time genomic surveillance, to quickly identify genetic loci which may be of public health interest. Further studies and analytical strategies in line with the present work can help improve real-time epidemiological surveillance in Brazil and the Americas, resulting in better public health policy outcomes.

We find signals in multiple genetic loci and present a structural-based review on the potential impact of changes at those loci.

However, the limited number of sequences analyzed demands caution when presenting the results. A large number of high-quality sequences is ideal for the application of ML analysis, especially when dealing with viruses, whose high mutation rates tend to insert many variations on its genomes. Furthermore, our analysis didn't consider virus-host interactions, such as host genome or immune system and pre-existing health conditions. In this regard, efficient host data collection, such as Electronic Health Records (EHR), are of paramount importance for a thorough investigation of clinical outcomes.

The envelope (E) protein is related to virus attachment and fusion [42]. NHP dataset analysis shows SNV L395F on Domain III, a region containing an IgC-like domain and supposed to contain a receptor-binding site crucial for virion maturation [43]. It is possible that this SNV could have an impact on the plasticity of this domain and affect the virus' receptor-binding site and it would be interesting to investigate this behaviour through simulations of molecular dynamics in future work.

NS1 protein is a crucial non-structural protein [36]. We found a non-synonymous SNV (A171V) on NHP dataset, located near a highly flexible region on the protein, called the *wing flexible loop*, which is a probable glycosylation site (GS) [44]. NS1 is also a key protein secreted by infected cells, which has the potential to interact with the adaptive immune system responses [36]. In dengue infections, NS1 is known to modulate capillary leakage in severe disease and may thus have a role to play in the severity of YFV infection [45, p.].

NS3 protein, which is composed of protease and helicase domains, has functions related to viral polyprotein processing and cleavage, viral genome replication and RNA capping [42]. SNV I184V, found on NHP dataset, is in a region with probable limited functional constraints [46]. SNV E614D, found on the human dataset, occurs in the NS3 helicase domain and is located in the RNA binding cleft. ASP and GLU are both negatively charged amino acids, but ASP has a shorter side chain which can cause it to lose access to the ligand. We used mCSM-NA [35] to evaluate the impact of the SNV on stability and affinity with RNA, showing a small destabilizing effect on the interaction with RNA (-0.646 Kcal/mol), which could have an impact on its function, fundamental to viral genome replication. Unfortunately, there are no structures in complex with RNA available. Models built with protein-RNA docking techniques could help elucidate if there is a significant impact of this SNV on RNA interaction.

We found one SNV on the NS4a protein (I69V, human dataset). However, a lack of current knowledge on YFV NS4a impeded us from further exploring the possible role of I69V in human hosts. Based on the protein's proposed functions [47–49], this SNV could, in principle, affect viral replication, but such hypothesis would have to be tested by non-computational means.

As an outlier, NS5 had the highest number of identified variations—I374V, E641D, N11S, R727G, V643A - from both human and NHP YFV sequences. NS5 protein is a fundamental enzyme for viral replication because it contains an N-terminal methyltransferase domain (MTase) and a C-terminal RNA dependent RNA polymerase domain (RdRp) [50]. MTase domain has important functions involved in protecting viral RNA from degradation and innate immunity response. RdRp is essential for viral RNA replication, because its activities cannot be performed by host polymerases, and is a promising target for antiviral drug development [51]. Furthermore, dengue virus NS5 has been associated with immune response evasion [52, p. 2]. With the structural analyses, we found that none of the SNVs found has been reported to be in positions with apparent connections to protein function or structure. However, R727G on the human dataset, on the thumb subdomain of RdRp domain, shows a change in predicted affinity for RNA upon SNV occurrence. This reduction in affinity could impact polymerase function and viral replication efficiency.

## Conclusions

In conclusion, even though the method proposed was applied on data that was already available from other sources, our study demonstrate that it is efficient and easy to replicate, making it suitable for real-time genomic surveillance, in which genetic data is analized as it is generated. This approach may help detect and inform on possible connections between ongoing genetic changes and public health in a timely manner.

## Materials and methods

### Ethics approval and consent to participate

This project was reviewed and approved by the Comissão Nacional de Ética em Pesquisa (CONEP) [National Research Ethics Committee] from the Brazilian Ministry of Health (BrMoH), as part of the arboviral genomic surveillance efforts within the terms of Resolution 510/2016 of CONEP, by the Pan American Health Organization Ethics Review Committee (PAHOERC) (Ref. No. PAHO-2016-08-0029), and by the Oswaldo Cruz Foundation Ethics Committee (CAAE: 90249218.6.1001.5248).

### Datasets

We retrieved YFV complete or near complete genome sequences from the recent Brazilian outbreaks, available on public databases [19–23], with associated epidemiological and clinical data containing relevant information regarding clinical severity and outcome for human infections samples, as well as PCR cycle threshold value (Ct) for both human and NHP samples. The alignment was made using MAFFT online [53] and was manually verified and corrected using AliView (https://ormbunkar.se/aliview/). Sequences with coverage lower than 90% were removed from the study. For ML analysis, human infection sequences were divided between not severe/not death and severe/death. *Callithrix* ssp. infection sequences were divided by low Ct ($<20$) and high Ct ($\geq 20$). After curation, the human dataset contained 56 YFV sequences, with 40 sequences related to severe/death cases, and 16 sequences related to not severe/not death cases. The NHP (*Callithrix* spp.) dataset, after curation, contained 27 sequences, of which 21 were related to low Ct values ($< 20$) and 6 were related to high Ct values ($> = 20$).

### Machine learning model adjustment

We applied three different ML models for each of two analyzed datasets, XGBoost [54, 55], random forest [56] and regularized logistic regression [57]. We adjusted the XGBoost model parameters in a "grid-search cross-validation" scheme with five folds. Random forest adjustment used "out of bag" data as validation. Regularized logistic regression parameters were adjusted on a 10-fold cross-validation scheme, using their averages in the final model. Part of the dataset (test set) was held out of model adjustment and validation, being used afterwards to test model's performance (as presented earlier).

### Model interpretation

Feature importance was computed using SHAP (SHapley Additive exPlanation) [30–32]. We followed the author's suggestion (https://github.com/slundberg/shap/issues/397) on dealing with categorical data when using SHAP.

### Protein structural analysis

We searched on Protein Data Bank (PDB) [58] for experimentally resolved structures. For those proteins that did not have structures, we looked for templates for comparative modeling with at least 30% identity. The comparative models were built with the Swiss-Model server [59]. We used mCSM [35] method to predict the impact of SNVs on protein stability and interactions.

## Author Contributions

**Conceptualization:** Álvaro Salgado, Adriano Veloso, José Lourenco, Luiz Carlos J. Alcantara.

**Data curation:** Álvaro Salgado, Marta Giovanetti, Talita Adelino.

**Formal analysis:** Álvaro Salgado, Raquel C. de Melo-Minardi.

**Funding acquisition:** Vasco Azevedo, Luiz Carlos J. Alcantara.

**Investigation:** Álvaro Salgado, Raquel C. de Melo-Minardi.

**Methodology:** Álvaro Salgado, Raquel C. de Melo-Minardi, Adriano Veloso.

**Project administration:** Marta Giovanetti, Vasco Azevedo, José Lourenco, Luiz Carlos J. Alcantara.

**Resources:** Vasco Azevedo, Luiz Carlos J. Alcantara.

**Software:** Álvaro Salgado, Francielly Morais-Rodrigues.

**Supervision:** Marta Giovanetti, Vasco Azevedo, José Lourenco, Luiz Carlos J. Alcantara.

**Validation:** Adriano Veloso, José Lourenco, Luiz Carlos J. Alcantara.

**Visualization:** Álvaro Salgado, Raquel C. de Melo-Minardi.

**Writing – original draft:** Álvaro Salgado, Raquel C. de Melo-Minardi.

**Writing – review & editing:** Marta Giovanetti, Ronaldo de Jesus, Stephane Tosta, José Lourenco, Luiz Carlos J. Alcantara.

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
