## [Decision Letter · Decision Letter 0]

17 Jun 2022

PONE-D-21-15059

Machine learning models exploring characteristic single-nucleotide signatures in yellow fever virus

PLOS ONE

Dear Dr. Salgado,

Thank you for submitting your manuscript to PLOS ONE. After careful consideration, we feel that it has merit but does not fully meet PLOS ONE’s publication criteria as it currently stands. Therefore, we invite you to submit a revised version of the manuscript that addresses the points raised during the review process.

We look forward to receiving your revised manuscript.

Kind regards,

Rakesh Kumar Verma, Ph.D

Academic Editor

PLOS ONE

Journal Requirements:

Reviewers' comments:

Reviewer's Responses to Questions

**Comments to the Author**

1. Is the manuscript technically sound, and do the data support the conclusions?

Reviewer #1: Yes

2. Has the statistical analysis been performed appropriately and rigorously? 

Reviewer #1: Yes

3. Have the authors made all data underlying the findings in their manuscript fully available?

Reviewer #1: Yes

4. Is the manuscript presented in an intelligible fashion and written in standard English?

Reviewer #1: No

5. Review Comments to the Author

Reviewer #1: The authors developed a machine learning framework combining three different algorithms to analyze YFV genomic sequences. This method was applied to 56 YFV sequences from human infections and 27 from non-human primate

(NHPs) infections and they investigated the presence of genetic signatures possibly related to disease severity (in human related sequences) and differences in PCR cycle threshold (Ct) values (in NHP related sequences).

The authors found four non-synonymous single nucleotide variations (SNVs) on sequences from human infections, in proteins NS3 (E614D), NS4a (I69V), NS5 (R727G, V643A) and six non-synonymous SNVs on NHP sequences, in proteins E (L385F), NS1 (A171V), NS3 (I184V) and NS5 (N11S, I374V, E641D).

They performed comparative protein structural analysis on these SNVs, trying to define possible impacts on protein function.

I have some major points to discuss here with authors:

Do you think using only Callithrix spp. sequences only as a NHP dateset can be a bias for your analysis since it know that Aluata spp. is the major sensitive NHP specimen for YFV in sylvatic cycle?

I think figure 2 needs to be more explained in Figure legend. It is hard for the readers understand the figure.

I think the authors should discuss better the importance of their findins in the control of YFV in Brazil and Latin America settings.

6. PLOS authors have the option to publish the peer review history of their article (what does this mean?). If published, this will include your full peer review and any attached files.

Reviewer #1: No

---

## [Author Response · Author response to Decision Letter 0]

4 Nov 2022

#1 Reviewer’s comment:

Do you think using only Callithrix spp. sequences only as a NHP dateset can be a bias for your analysis since it know that Aluata spp. is the major sensitive NHP specimen for YFV in sylvatic cycle?

Author’s answer:

The use of only Callithrix spp. sequences was intentional. It is indeed known that Aluata spp. is the major sensitive NHP specimen for YFV in sylvatic cycle, but these animals usually live far from human populations, in forest areas separated from large urban centers, and are unlikely to be responsible for starting an urban cycle of transmission. On the other hand, Callithrix spp. animals are easily found in densely populated urban areas, in parks and in forested regions close to cities. That is why a variation in PCR cycle threshold (Ct) values constitutes a greater risk in igniting an urban cycle of transmission, because it can signify a change in persistence and infectivity of Yellow Fever Virus in Callithrix spp.

Therefore, the present work intends to investigate if there are any genetic signatures possibly related to this specific phenomenon, and this is why the work focused on Callithrix spp. alone.

#2 Reviewer’s comment:

I think figure 2 needs to be more explained in Figure legend. It is hard for the readers understand the figure.

Author’s answer:

Change was made on manuscript, replicated here for your convenience:

“Machine learning models’ performance

Fig 2 shows the confusion matrices for the machine learning models applied. For the human dataset, the XGBoost classification model correctly classified 16 serious/death and 5 not serious/not death cases, out of a total of 28 instances, achieving an accuracy of 75% on the test set, with F-1 scores of 0.59 and 0.82 for classes 0 (not severe/not death) and 1 (severe/death), respectively. The random forest model correctly classified 16 serious/death and 7 not serious/not death cases, out of a total of 28 instances, achieving an accuracy of 82% on the test set, with F-1 scores of 0.74 and 0.86 for classes 0 and 1 respectively. The modified logistic regression model performance was the same as that of random forest. 

For the Callithrix spp. dataset, the three models achieved an accuracy of 100% on the test set, correctly predicting 5 low Ct cases and 1 high Ct case, out of 6 instances in the test set, with F-1 scores of 1.00 and 1.00 for classes 0 (low Ct) and 1 (high Ct) respectively. 

Fig 2. Confusion matrices. Each box contains one confusion matrix, which measures the performance of different machine learning algoritms over different datasets. For each matrix, the rows represent instances in the actual class, while the columns represent instances in the predicted class. Instances found along the diagonal were correctly classified, while those outside the diagonal were misclassified (false positives and false negatives). Top three matrices correspond to human test dataset and the bottom three matrices correspond to Callithrix spp. test dataset, for XGBoost, random forest and regularized logistic regression respectively.

”

#3 Reviewer’s comment:

I think the authors should discuss better the importance of their findins in the control of YFV in Brazil and Latin America settings.

Author’s answer:

Changes were made on manuscript, replicated here for your convenience:

“Emerging and reemerging viruses present a highly complex challenge for the Brazilian public health system. Among them, arboviruses transmitted by mosquitoes are agents capable of causing serious diseases, such as hemorrhagic fevers, encephalitis and meningitis. For these reasons, real-time genomic surveillance is extremely important to guide prevention and control measures, as it allows reconstruction of the origins of epidemics and the estimation of transmission rates at different times and geographic regions, subject to environmental and human factors. In addition, genomic surveillance makes it possible to identify emerging, re-emerging, circulating and co-circulating variants, through viral genetic diversity quantification, making it possible to estimate the likelihood of new outbreaks and/or possible escapes from existing vaccines and treatments. As a result, relevant information is acquired for the design of public health policies, in addition to contributing to the development of vaccines, new drugs and improved serological and molecular diagnostic methods (40,41).

In this context, Brazil has become a global reference in real-time genomic surveillance, achieving fundamental results in early detection and monitoring of outbreaks. However, the large amount of data produced using next-generation sequencing platforms demands sophisticated analytical approaches, capable of dealing with complex and large datasets, aiming at the the extraction of as much information as possible. In this sense, Machine Learning algorithms have been successfully used in Bioinformatics, motivating their application in the search for genetic signatures in arboviruses, associated with phenotypic or epidemiological characteristics in recent outbreaks in Brazil.

In this study, we demonstrate the potential of applying ML approaches on real-time genomic surveillance, to quickly identify genetic loci which may be of public health interest. Further studies and analytical strategies in line with the present work can help improve real-time epidemiological surveillance in Brazil and the Americas, resulting in better public health policy outcomes.

”

---

## [Editor Report · Decision Letter 1]

29 Nov 2022

Machine learning models exploring characteristic single-nucleotide signatures in yellow fever virus

PONE-D-21-15059R1

Dear Dr. Salgado,

We’re pleased to inform you that your manuscript has been judged scientifically suitable for publication and will be formally accepted for publication once it meets all outstanding technical requirements.

Kind regards,

Rakesh Kumar Verma, Ph.D

Academic Editor

PLOS ONE

Additional Editor Comments (optional):

Dear Author,

The Manuscript fulfil all the requirements and finds suitable for publication in PLOS-ONE Journal.

Thank You
---

## [Editor Report · Acceptance letter]

3 Dec 2022

PONE-D-21-15059R1 

Machine learning models exploring characteristic single-nucleotide signatures in yellow fever virus 

Dear Dr. Salgado:

I'm pleased to inform you that your manuscript has been deemed suitable for publication in PLOS ONE. Congratulations! Your manuscript is now with our production department. 

Kind regards, 

on behalf of

Dr. Rakesh Kumar Verma 

Academic Editor

PLOS ONE